Classifying reservoir facies using attention-based residual neural networks

Nguyen An Hai 1 annh1@pvep.com.vn
Nguyen Khang 2
Mai Nga 3
1 R&D Department, Petrovietnam Exploration Production Corporation , Hanoi , Vietnam
2 IBM Vietnam , Hanoi , Vietnam
3 Phenikaa School of Computing, Phenikaa University , Hanoi , Vietnam
Nguyen Hoang
Electronic publication date: 2025 Jul 8
Publication date: 2025
Volume: 11
Electronic Location ID: e2977
Received 2025 Feb 17; Accepted 2025 May 30
Copyright: © 2025 Nguyen et al.
Copyright year: 2025
Copyright holder: Nguyen et al.
License: This is an open access article distributed under the terms of the Creative Commons Attribution License, which permits unrestricted use, distribution, reproduction and adaptation in any medium and for any purpose provided that it is properly attributed. For attribution, the original author(s), title, publication source (PeerJ Computer Science) and either DOI or URL of the article must be cited.
License URL: https://creativecommons.org/licenses/by/4.0/

Keywords: Reservoir, Facies, Attention, Deep learning, Machine learning

Funding: The authors received no funding for this work.

==============================
The accurate classification of reservoir facies remains a fundamental challenge in petroleum geoscience, with significant implications for resource extraction efficiency and reservoir characterization. Traditional approaches relying on manual interpretation and conventional machine learning methods often struggle with the complexity and heterogeneity of well-log data. This architectural approach, in contrast to traditional single-stream or non-residual designs, significantly enhances the model’s ability to concentrate on key geological features while preserving hierarchical representations of the data. Consequently, it more effectively addresses data heterogeneity and contextual dependencies. The framework was trained and evaluated using measurements from eight wells that represent diverse geological settings. Comparative experiments against conventional machine learning models and state-of-the-art deep learning techniques demonstrated the superiority of our method, achieving an area under the receiver operating characteristic curve (AUROC) of 0.883 and an area under the precision-recall curve (AUPRC) of 0.502. These enhancements enable more accurate identification of subtle facies boundaries and lithological variations, particularly in complex geological formations, thereby facilitating improved reservoir delineation and reducing uncertainty in field development planning. Furthermore, reproducibility analyses confirmed consistent performance across independent trials, underscoring the model’s robustness and its viability for real-world reservoir characterization workflows.

Introduction

In sedimentology, a facies is defined as a body of rock with distinct characteristics that differentiate it from other rock bodies. Based on differences in lithological, biological, and sedimentary structures, facies can be classified to reflect their depositional environments and the processes that created them (Armas & Sánchez, 2015; Garcia et al., 2021). This classification is essential for understanding Earth’s geological history, revealing how sediments were formed and deposited, and understanding rock distribution within formations (Beatriz Soto, Leonardo Durán & Aldana, 2014; Moreau et al., 2023). Suter (2006) emphasized the importance of facies models in leveraging advanced technologies across various depositional environments, particularly in clastic environments such as deltaic and deep-water settings. Well-logs represent the primary source of information for facies classification and provide indispensable data about subsurface oil and gas reservoirs. By recording the electrical responses and nuclear radiation of drilled sections, well-logs enable inference of rock matrix and fluid properties. This information supports applications ranging from exploration and development to production forecasting and monitoring. However, the values recorded by logging instruments do not directly reflect formation characteristics and must undergo processing to derive reservoir parameters. While geophysical logging is grounded in robust mathematical principles, geological complexity often exceeds the capabilities of conventional mechanistic models (Xie et al., 2018).

Lithology, referring to the physical and mineralogical characteristics of rocks, is a crucial factor in rock type classification. It encompasses composition, grain size, texture, and color. Lithology identification can be achieved through two primary methods: direct observations (including core samples and mud logs) and indirect geophysical methods that rely on correlations between lithologies and their physical and elastic properties, inferred from logging curves such as gamma ray, resistivity, sonic, density, and neutron (Serra & Abbott, 1982). However, field-acquired data are often highly sampled and voluminous, creating significant interpretation challenges (Horrocks, Holden & Wedge, 2015). Several challenges exist in lithology interpretation. Core sampling methods, while providing direct measurements, may yield varying interpretations and involve costly, time-intensive operations (Maia Ramos Lopes & Neves Andrade, 2019). Additionally, interpreting lithology using logging curves presents a non-linear problem, as the combined effects of mineral composition, cementation, and fluid types influence logging responses (Onyekuru et al., 2021). Consequently, implementing suitable non-linear mapping techniques is essential for automated and intelligent rock type identification (Li et al., 2023).

Accurately classifying facies is crucial for enhancing the efficiency of resource extraction, which is highly essential for energy security and economic development. Traditionally, facies classification has primarily relied on qualitative assessments through visual examination of rock samples, stratigraphic relationships, and sedimentary structures. Miall (1976) investigated the importance of sedimentary structures and lithology in defining facies types. Miall’s work established the foundation for understanding how sedimentary processes control facies distribution, particularly in fluvial and floodplain environments (Burgess, 2016). The evolution of facies classification methodologies reflects significant technological advancements. Early statistical approaches, such as Soltanian & Ritzi’s (2014) clustering method for lithofacies identification in the Zamzama gas field, represented a critical transition from purely visual interpretations. Traditional machine learning (ML) techniques, like support vector machine (SVM) introduced by Hall (2016), demonstrated initial promise by applying algorithmic approaches to well-log measurements. However, these methods faced substantial limitations: they typically required extensive feature engineering, struggled with complex non-linear relationships, and often relied on manually curated geological features with limited generalizability.

Deep learning (DL) approaches emerged as a transformative solution, addressing many constraints of traditional ML methods. Unlike older approaches that required explicit feature extraction, DL models can autonomously learn hierarchical representations directly from raw data. Zhao’s (2018) review highlighted the paradigm shift, demonstrating DL’s capacity to handle intricate geological patterns that previous methods could not effectively capture. Notably, advanced architectures like convolutional neural networks (CNNs) and autoencoders, as demonstrated by Liu et al. (2021), can automatically extract complex features, reducing human intervention and potentially uncovering subtle geological signatures previously overlooked. Recent developments, such as Grana, Azevedo & Liu’s (2020) integration of DL with probabilistic methods and Feng et al.’s (2021) Bayesian convolutional neural networks, further underscore the field’s progression. These approaches not only improve classification accuracy but also provide uncertainty quantification, a critical aspect often missing in traditional geological interpretation methods. By leveraging sophisticated neural network architectures, researchers can now process multidimensional geological data with unprecedented complexity and precision, marking a significant leap from earlier computational approaches to facies classification.

While DL approaches often surpass classical ML models in representational power and classification accuracy, they commonly face challenges related to interpretability, generalization across different wells, and the management of mixed data types-particularly the integration of categorical and numerical well-log features within a cohesive framework. Models based on conventional convolutional neural networks and recurrent neural networks tend to assume spatial or sequential correlations, which may not fully capture the contextual relationships inherent in well-log data. To address these challenges, recent research has investigated hybrid architectures that leverage attention mechanisms and residual connections. Despite these advancements, a notable gap remains in the development of models that effectively integrate residual learning and attention mechanisms within a structured architecture specifically designed for heterogeneous well-log inputs.

Although various DL models have been developed for facies classification, existing approaches may not fully leverage the inherent features present in the data. To address this limitation, we propose an more effective DL architecture that integrates the Transformer (Vaswani et al., 2017) with a deep residual network (ResNet) (He et al., 2016). This combination enhances the model’s capacity to capture distinctive characteristics in the data. Unlike traditional DL architectures that primarily excel at processing numerical features (Vaswani et al., 2017), specialized Transformer variants are designed to effectively handle both numerical and categorical features through their modified attention mechanisms (Huang et al., 2020; Badaro, Saeed & Papotti, 2023; Mao, 2024). These tabular data-adapted Transformers can therefore capture complex relationships and contextual dependencies among features, independent of sequential order. Complementing the Transformer’s capabilities, ResNet architecture demonstrates superior performance in processing numerical features through its distinctive structure. The network implements residual connections between layers, facilitating direct information flow from input to output, thereby addressing the vanishing gradient problem prevalent in DL models (He et al., 2016). By combining the strengths of both architectures, our proposed framework integrates their respective advantages to elevate the performance of facies classification, providing a robust solution for this challenging task. Notably, this approach offers enhanced interpretability through attention visualization techniques, enabling geoscience experts to understand how the model derives its classifications by revealing critical feature interactions and subsurface signal weightings across different geological depths.

Data collection

To develop our model, we used a dataset from Hall’s (2016) study on rock facies classification. The dataset contains 3,232 well-log entries from eight wells in the Hugoton field of southwest Kansas (Dubois, Bohling & Chakrabarti, 2007). Each entry includes five wireline log measurements, two indicator variables, measurement depth, and a classification into one of nine facies categories. The facies’s labels and their descriptions are provided in Table 1. We used a pairwise plot to provide insights into how rock properties vary across facies and to explore relationships between numerical variables (Fig. 1). Since the dataset contains five log variables, a scatter matrix can effectively display their interdependencies.

Table 1 Description of facies’ labels.

Facies’ label	Description	
1	Nomarine sandstone (SS)	
2	Nonmarine coarse siltstone (CSiS)	
3	Nonmarine fine siltstone (FSiS)	
4	Marine siltstone and shale (SiSh)	
5	Mudstone (MS)	
6	Wackestone (WS)	
7	Dolomite (D)	
8	Packstone-grainstone (PS)	
9	Phylloid-algal bafflestone (BS)	

Figure 1 Pairwise analysis on features of all well logs.

We conducted eight separate experiments, each using data from one well as the test set while using the remaining wells’ data for training and validation in a 9:1 ratio. Table 2 summarizes the distribution of samples across these eight experimental configurations. The facies analysis was performed using core samples taken at half-foot intervals, which were correlated with logging data from the respective well locations. The feature variables include GR (Gamma Ray), ILD_log10 (Deep Induction Log), PE (Photoelectric Effect), DeltaPHI (Neutron-Density Porosity Difference), PNHIND, and NM_M. Figure 2 illustrates these measurements using data from the SHRIMPLIN well as an example.

Table 2 The distribution of samples across these eight experimental configurations.

Well type	Training	Validation	Test	
CHURCHMAN BIBLE	3,370	375	404	
LUKE G U	3,319	369	461	
CROSS H CATTLE	3,283	365	501	
Recruit F9	3,662	407	80	
NOLAN	3,360	374	415	
NEWBY	3,317	369	463	
SHRIMPLIN	3,310	368	471	
SHANKLE	3,330	370	449	

Figure 2 Well log measurements from the SHRIMPLIN well.

Model architecture

Our architecture processes two types of inputs: categorical and numerical features, through parallel processing streams that are later combined. For categorical features, the data first passes through an embedding layer, followed by a Transformer block (Vaswani et al., 2017). The Transformer consists of two main components: a multi-head self-attention mechanism and a feed-forward network, each followed by an Add & Norm layer for residual connections and normalization. For numerical features, we implement a Residual block that begins with a Normalization layer, followed by two sequential blocks of fully connected (FC) layers (dimensions 7 × 64 and 64 × 64), batch normalization (BatchNorm) (Ioffe & Szegedy, 2015), and gated linear unit (GLU) activations. The outputs from both streams are merged through concatenation and fed into a final multilayer perceptron (MLP) (Haykin, 1998) for prediction. This dual-stream architecture, illustrated in Fig. 3, effectively handles both categorical and numerical features while maintaining the ability to capture complex patterns in the data.

Figure 3 Model architecture.

Transformer

A Transformer consists of a multi-head self-attention layer followed by a position-wise feed-forward layer, with element-wise addition and layer normalization being done after each layer (Vaswani et al., 2017). A self-attention layer comprises three parametric matrices Key, Query, and Value. Each input embedding is projected onto these matrices, to generate their Key, Query, and Value vectors. Formally, let K∈Rm×k,Q∈Rm×k and V∈Rm×v be the matrices comprising key, query and value vectors of all the embeddings, respectively, and m be the number of embeddings inputted to the Transformer, k and v be the dimensions of the key and value vectors, respectively. Every input embedding attends to all other embeddings through an Attention head, which is computed as follows:

(1) Attention(K,Q,V)=softmax(QKTk)⋅V.

This operation allows each feature to attend to all other features, weighting their contributions based on relevance, thereby capturing both short- and long-range dependencies. In well-log data, where features such as resistivity, porosity, and gamma-ray measurements interact in complex, non-sequential ways, this attention mechanism facilitates feature interaction modeling without relying on strict positional order. We use multi-head attention, which allows the model to learn multiple, complementary relationships between features in parallel. The output is passed through a feed-forward network (FFN) consisting of two linear layers with ReLU activation, followed by residual addition and layer normalization:

(2) FFN(x)=ReLU(xW1+b1)⋅W2+b2.

The residual structure and normalization facilitate gradient flow and prevent feature drift during training, enhancing model robustness.

Residual block

The Residual block is designed to learn effective and robust numerical features, taking inspiration from the TabNet architecture (Arik & Pfister, 2019) for tabular data processing (Arik & Pfister, 2019). The block consists of two sequential layers, where each FC layer is followed by a BatchNorm and a GLU activation function (Dauphin et al., 2016). A residual connection combines the input with the final output after the GLU layer, helping to stabilize the learning process by maintaining consistent variance throughout the network (Gehring et al., 2017). The first FC layer expands the feature dimension from 7 to 64, while the second FC layer maintains the 64-dimensional representation with a 64×64 transformation. The detailed architecture is as follows: Input dimension: R7 (e.g., 7 numerical well-log features)

FC1: Expands the input to R64

BatchNorm1 → GLU1 (outputs R64)

FC2: R64×64 transformation

BatchNorm2 → GLU2

A residual connection adds the original input (after projection if needed) to the output of the second GLU activation:

(3) Output=GLU2(BN2(FC2(GLU1(BN1(FC1(x))))))+x.

The use of residual connections in the network enhances gradient stability and ensures a consistent feature scale across various layers, simplifying the training of deep networks. Incorporating BatchNorm layers helps reduce internal covariate shift, speeding up convergence and making the network less sensitive to the initial settings of its parameters. Furthermore, implementing GLUs instead of traditional activation functions like ReLU has proven effective for structured data. This method offers improved gating dynamics and feature sparsity, ultimately leading to better interpretability and generalization of the model. Together, these design choices enable the Residual block to serve as a stable and expressive numerical encoder in the overall dual-stream architecture.

Feature representation

Let (x,y) denote a feature-target pair, where x=(xcat,xcont). Here, xcat represents all categorical features and xcont∈R represents continuous features. Let xcat=(x1,x2,x3,⋯,xm), where each xi is a categorical feature for i∈1,⋯,m. Each categorical feature xi is encoded into a vector with dimension d using a Transformer layer. Let Tcat∈Rd for i∈1,⋯,m represent the encoded features obtained after the Transformer block, computed as:

(4) Tcat=T(xcat)=(T(x0),T(x1),T(x2),⋯,T(xm)),

where Tcat is the set of embeddings for all categorical features and T denotes the operation of the Transformer block. The numerical features xi are processed by a normalization layer before being forwarded to the residual block as follows:

(5) Rcont=(R(h0),R(h1),R(h2),⋯,R(hn)),

where hi∈R represents the continuous features, R denotes the normalization process, and Rcont∈Rn is the set of encoded features obtained after the Residual block for i∈1,⋯,n. The encoded categorical and numerical vectors are then concatenated into a unified vector Fconcat with a dimension of 128, described as:

(6) Fconcat=(TcatRcont)=(x1x2⋮xmh1h2⋮hn)

This vector is finally passed to the MLP layer, denoted by Gψ, to predict the target y. Cross-entropy is used as the loss function for the classification tasks.

Experiments

Experimental setup

Our models were trained for 50 epochs using a learning rate of 0.01, a batch size of 64, and the Adam optimizer (Kingma & Ba, 2014). The experiments were conducted using PyTorch framework version 2.0.0. The DL models were executed on an RTX 3090 GPU with 24 GB of memory. Data loading and processing were performed on a computer equipped with an AMD Ryzen 7 5800X 8-Core Processor (3.80 GHz) running Windows 11 Professional.

Data preprocessing

The dataset comprised well-log records from eight wells, representing a range of geological settings. Each record included both numerical features (such as gamma-ray, resistivity, and porosity) and categorical attributes (like lithology indicators). Numerical features were standardized using z-score normalization to achieve a mean of zero and a variance of one across all features. Categorical variables were transformed into dense vector representations through learnable embedding layers optimized jointly during training. Missing values in the numerical features were imputed using the mean of each feature while missing categorical values were represented by a dedicated ‘unknown’ embedding token.

Evaluation metrics

Model performance was evaluated using two primary metrics: Area under the receiver operating characteristic curve (AUROC): This metric evaluates the model’s ability to distinguish between facies classes at different decision thresholds. It is particularly effective for assessing classification performance when class distributions are balanced or moderately imbalanced.

Area under the precision-recall curve (AUPRC): This metric assesses the balance between precision and recall, making it particularly useful in situations of class imbalance. Given that some facies types are underrepresented in the dataset, AUPRC offers a more accurate measure of the model’s ability to identify minority classes.

These metrics and accuracy (ACC) and F1-score (F1) were calculated on the held-out test set. The final results reported represent the average performance across five independent training runs, with 95% confidence intervals obtained through bootstrap resampling to ensure statistical reliability.

Hyperparameter tuning

Table 3 provides details of hyperparameter configurations for our proposed model, systematically exploring key architectural components across multiple dimensions. For the embedding layer, we investigated dimensions ranging from 16 to 64, with 32 selected as the default setting. The Transformer block underwent extensive tuning, examining the number of attention heads (2–4), Transformer layer configurations (2–4 layers), and hidden feed-forward network dimensions (64–256). Similarly, the residual block’s fully connected layers were explored across dimensions of 32–128, allowing flexibility in feature representation. The multilayer perceptron (MLP) layer’s hidden dimension was varied between 64 and 256, with 128 chosen as the default. This methodical approach enables a structured exploration of the model’s architectural hyperparameters, balancing computational efficiency with potential performance gains by systematically varying key network components. Ultimately, the best hyperparameters were selected based on the evaluated performance of the model on the validation set.

Table 3 Hyperparameter tuning settings for our models.

Component	Hyperparameter range	Default value	Description	
Embedding layer	[16, 32, 64]	32	Embedding dimension	
Transformer block	[2, 4]	4	Number of attention heads	
[2, 3, 4]	2	Number of transformer layers	
[64,128, 256]	64	Hidden dimension (FFN)	
Residual block	[32, 64, 128]	64	Dimension of the first FC layer’s output	
[32, 64, 128]	64	Dimension of the second FC layer’s output	
MLP layer	[64, 128, 256]	128	Hidden layer	

The proposed model and all baseline models were trained for up to 50 epochs using the Adam optimizer (Kingma & Ba, 2014) with a learning rate of 0.01. This learning rate was determined through a grid search over the following values: 0.001, 0.005, 0.01, 0.05, with validation performance guiding the selection. A batch size of 128 was used for all experiments. To prevent overfitting, dropout was applied with a rate of 0.3, and L2 weight regularization was also utilized. Early stopping based on validation, AUPRC was implemented with a patience of 10 epochs. Each experiment was repeated five times with different random seeds to assess consistency, and the best-performing model on the validation set was chosen for testing. All model parameters, architectural configurations, and tuning strategies were kept consistent across all comparisons to ensure a fair evaluation.

Comparison with baseline models

The performance of our proposed model was evaluated against several established ML and DL models across eight well types. The comparative analysis incorporated multiple ML models: SVM (Alvarsson et al., 2016) for their optimal hyperplane classification capabilities; Gaussian process classifier (GPC) Villacampa-Calvo et al. (2020) utilizing probabilistic function distributions; and Random Forest classifier (RFC) (Breiman, 2001) leveraging ensemble-based decision tree integration. Tree-based methods were further represented by the eXtreme Gradient Boosting (XGB) (Chen & Guestrin, 2016; Deng et al., 2021) model, which implements sequential tree construction with error correction mechanisms, and the fundamental decision tree (DT) (Wu et al., 2007) model based on feature-driven decision paths. The instance-based k-nearest neighbors ( k-NN) (Mucherino, Papajorgji & Pardalos, 2009) model was included for its proximity-based classification methodology. For DL approaches, we used the artificial neural network (ANN) (McCulloch & Pitts, 1943), characterized by its neural layered architecture, and the CNN (O’Shea & Nash, 2015), distinguished by its hierarchical feature learning through convolution operations.

Comparison with deep learning models

To conduct a comprehensive performance analysis, we also compared our proposed model with several advanced DL methods that have been implemented on tabular data. These methods include DCN-V2 (Wang et al., 2020), an architecture for learning-to-rank that uses cross-network and self-network components to efficiently capture feature interactions, improving upon the original deep and cross network (DCN) model. TabNet (Arik & Pfister, 2019) is a deep tabular data learning architecture that applies sequential attention to choose which features to reason from at each decision step, enabling interpretability. FT-Transformer (Feature Tokenizer + Transformer) is a simple adaptation of the Transformer architecture, originally designed for language modeling, to the tabular domain by tokenizing features (Gorishniy et al., 2021). TabNN (Ke et al., 2019) is a universal neural network solution that automatically derives effective architectures customized to specific tabular datasets and predictive modeling tasks.

Results and discussion

Comparison with baseline models

Table 4 compares the performance of the proposed model against several baseline ML and DL models across eight well types. It is evident that the proposed model consistently outperforms the other models in terms of both AUROC and AUPRC across all well types. The proposed model achieves the highest AUROC and AUPRC values for each well type, demonstrating its superior classification performance compared to the baseline models. Among the baseline models, XGB and k-NN generally perform better than the other ML models, with XGB showing the highest AUROC and AUPRC values for most well types. The DL models, ANN and CNN, also demonstrate competitive performance, often surpassing the ML models but falling short of the proposed model’s performance.

Table 4 Performance comparison between ours and other baseline models.

Well type	Metric	Model	
		Ours	SVM	RFC	XGB	KNN	DT	ANN	CNN	CatBoost	
SHRIMPLIN	AUROC	0.908	0.889	0.891	0.870	0.815	0.708	0.857	0.866	0.880	
AUPRC	0.580	0.523	0.574	0.450	0.423	0.305	0.423	0.460	0.510	
ACC	0.552	0.544	0.554	0.529	0.535	0.486	0.529	0.558	0.545	
F1	0.368	0.393	0.425	0.384	0.369	0.379	0.372	0.401	0.395	
SHAKLE	AUROC	0.882	0.835	0.868	0.844	0.743	0.649	0.817	0.852	0.860	
AUPRC	0.480	0.381	0.459	0.427	0.335	0.253	0.423	0.432	0.440	
ACC	0.535	0.461	0.459	0.454	0.457	0.370	0.448	0.503	0.510	
F1	0.417	0.315	0.336	0.347	0.329	0.320	0.368	0.388	0.380	
NOLAN	AUROC	0.864	0.791	0.829	0.844	0.703	0.613	0.858	0.853	0.835	
AUPRC	0.418	0.351	0.366	0.406	0.316	0.227	0.402	0.393	0.395	
ACC	0.460	0.443	0.431	0.484	0.446	0.448	0.463	0.521	0.480	
F1	0.276	0.253	0.248	0.299	0.277	0.263	0.279	0.312	0.320	
NEWBY	AUROC	0.878	0.823	0.846	0.854	0.749	0.617	0.877	0.859	0.850	
AUPRC	0.552	0.399	0.456	0.482	0.352	0.225	0.569	0.497	0.510	
ACC	0.477	0.391	0.445	0.436	0.417	0.372	0.546	0.449	0.470	
F1	0.382	0.285	0.327	0.375	0.317	0.287	0.492	0.411	0.410	
LUKA GU	AUROC	0.924	0.879	0.898	0.898	0.825	0.644	0.891	0.904	0.915	
AUPRC	0.575	0.464	0.581	0.544	0.480	0.288	0.500	0.513	0.530	
ACC	0.525	0.451	0.586	0.514	0.475	0.471	0.518	0.464	0.500	
F1	0.388	0.293	0.411	0.388	0.353	0.301	0.379	0.357	0.385	
CROSS H CATTLE	AUROC	0.841	0.810	0.847	0.822	0.745	0.631	0.786	0.843	0.825	
AUPRC	0.407	0.373	0.436	0.388	0.344	0.224	0.396	0.401	0.395	
ACC	0.355	0.317	0.383	0.363	0.379	0.357	0.341	0.429	0.420	
F1	0.316	0.239	0.343	0.263	0.309	0.273	0.293	0.322	0.340	
CHURCHMAN BIBLE	AUROC	0.886	0.836	0.877	0.842	0.763	0.682	0.851	0.856	0.87	
AUPRC	0.488	0.446	0.473	0.412	0.359	0.274	0.441	0.429	0.450	
ACC	0.527	0.475	0.530	0.498	0.490	0.480	0.537	0.483	0.490	
F1	0.387	0.389	0.411	0.388	0.366	0.396	0.416	0.393	0.405	
Note:

Bold values represent the best statistically significant results.

The performance gap between the proposed model and the baseline models varies across well types. In the SHRIMPLIN well type, the proposed model achieves an AUROC of 0.908 and an AUPRC of 0.580, significantly outperforming the best baseline model, XGB, which has an AUROC of 0.870 and an AUPRC of 0.450. In contrast, for the NOLAN well type, the performance gap is smaller, with the proposed model achieving an AUROC of 0.864 and an AUPRC of 0.418, while the best baseline model, XGB, has an AUROC of 0.844 and an AUPRC of 0.406. Overall, the results presented in the table demonstrate the effectiveness of the proposed model in classifying various well types, showcasing its superior performance compared to established ML and DL baseline models.

Comparison with deep learning models

The experimental results demonstrate compelling evidence of our model’s superior performance across multiple well types and evaluation metrics compared to existing state-of-the-art approaches (Table 5). In terms of AUROC values, our model consistently outperforms all competitors, with particularly impressive results in LUKA GU (0.924) and SHRIMPLIN (0.908) wells, representing substantial improvements over the second-best performer, the FT-Transformer. The AUPRC values further underscore our model’s capabilities, maintaining significant leads across all well types, notably in SHRIMPLIN (0.580), LUKA GU (0.575), and NEWBY (0.552) wells. Among competing models, the FT-Transformer emerges as the consistent second-best performer, particularly in AUROC values, while the baseline DCN model shows the weakest performance across all categories. The consistent performance hierarchy across different well types suggests that our model’s superior results represent a fundamental improvement in the underlying modeling approach rather than being limited to specific well characteristics. These results strongly indicate that our proposed architecture successfully addresses key limitations of existing approaches, particularly in handling complex well data characteristics and class imbalance issues.

Table 5 Performance comparison between ours and other state-of-the-art models.

Well type	Metric	Model	
		Ours	TabNet	FT-Trans	TabNN	DCN	TabTrans	TabFPN	
SHRIMPLIN	AUROC	0.908	0.780	0.900	0.834	0.706	0.890	0.875	
AUPRC	0.580	0.383	0.440	0.456	0.296	0.420	0.400	
ACC	0.552	0.522	0.512	0.524	0.501	0.500	0.495	
F1	0.368	0.368	0.289	0.364	0.391	0.270	0.260	
SHAKLE	AUROC	0.882	0.723	0.883	0.778	0.645	0.870	0.860	
AUPRC	0.480	0.313	0.473	0.385	0.244	0.450	0.430	
ACC	0.535	0.408	0.535	0.443	0.367	0.525	0.520	
F1	0.417	0.332	0.379	0.373	0.319	0.360	0.350	
NOLAN	AUROC	0.864	0.700	0.860	0.730	0.609	0.850	0.840	
AUPRC	0.418	0.319	0.359	0.339	0.232	0.340	0.320	
ACC	0.460	0.446	0.504	0.455	0.448	0.490	0.480	
F1	0.276	0.329	0.310	0.349	0.354	0.290	0.280	
NEWBY	AUROC	0.878	0.739	0.829	0.786	0.617	0.810	0.800	
AUPRC	0.552	0.349	0.405	0.394	0.223	0.380	0.360	
ACC	0.477	0.408	0.458	0.443	0.365	0.450	0.440	
F1	0.382	0.365	0.244	0.375	0.326	0.250	0.240	
LUKA GU	AUC	0.924	0.832	0.887	0.838	0.630	0.870	0.860	
PR_AUC	0.575	0.488	0.523	0.494	0.267	0.500	0.480	
ACC	0.525	0.488	0.605	0.488	0.436	0.570	0.560	
F1	0.388	0.438	0.287	0.433	0.324	0.270	0.260	
CROSS H CATTLE	AUROC	0.841	0.729	0.837	0.791	0.669	0.820	0.810	
AUPRC	0.407	0.302	0.369	0.376	0.240	0.350	0.340	
ACC	0.355	0.339	0.329	0.369	0.365	0.340	0.330	
F1	0.316	0.369	0.346	0.385	0.343	0.320	0.310	
CHURCHMAN BIBLE	AUROC	0.886	0.781	0.833	0.803	0.677	0.820	0.810	
AUPRC	0.488	0.378	0.371	0.388	0.272	0.350	0.340	
ACC	0.527	0.470	0.428	0.498	0.485	0.420	0.410	
F1	0.387	0.445	0.227	0.382	0.455	0.230	0.220	
Note:

Bold values represent the best statistically significant results.

Modeling reproducibility

The modeling reproducibility analysis of our proposed method demonstrates robust and consistent performance across multiple trials (Table 6). Mean, standard deviation (SD), and 95% confidence intervals (95% CI) were computed investigate the performance variation of the method. The model achieved an AUROC value of 0.883 ± 0.027 and an AUPRC of 0.502 ± 0.071, with 95% CI of [0.863–0.903] and [0.449–0.555], respectively. Notable performance peaks were observed in trial 5, which recorded the highest AUROC of 0.924 and AUPRC of 0.575, while trial six showed relatively lower metrics with an AUROC of 0.841 and AUPRC of 0.407. The narrow SD in AUROC value suggests stable model performance across different trials, though the slightly higher variability in AUPRC values indicates some sensitivity to class imbalance in the dataset. The small CI for both metrics further support the reliability of the model’s performance, providing strong statistical evidence for the reproducibility of our methodology.

Table 6 Modeling reproducibility of proposed method.

Trial	AUROC	AUPRC	
1	0.908	0.58	
2	0.882	0.48	
3	0.864	0.418	
4	0.878	0.552	
5	0.924	0.575	
6	0.841	0.407	
7	0.886	0.488	
Mean	0.883	0.502	
SD	0.027	0.071	
95% CI	[0.863–0.903]	[0.449–0.555]	

Limitations

Despite promising outcomes, our method has shortcomings that need to be addressed in the future. The model’s reliance on high-quality labeled training data may limit its applicability in regions with sparse or uncertain facies annotations. While the dual-stream architecture enhances feature processing, it introduces additional computational overhead compared to simpler approaches, potentially affecting deployment in systems with limited computing resources. The current implementation assumes consistent well-log measurement types across all wells, which may not hold in cases involving legacy data or varying logging tool specifications. The model’s performance shows some sensitivity to class imbalance, as evidenced by the variability in AUPRC values across different facies types. In terms of applicability, the proposed method demonstrates promising performance in facies classification, but its practical implementation is constrained by requirements for high-quality labeled training data, consistent well-log measurement types, and computational resources.

Conclusion

In our study, we developed and evaluated an effective dual-stream neural network that had outperformed the state-of-the-art in reservoir facies classification. The framework demonstrated exceptional performance, achieving mean AUROC of 0.883 and AUPRC of 0.502 across diverse well types, significantly higher than those of both traditional ML and contemporary DL models. The model’s stability was confirmed through comprehensive reproducibility analysis, with consistent performance across multiple trials as evidenced by narrow confidence intervals. Our model demonstrated effective handling of heterogeneous well-log data and robust performance in geologically complex scenarios. Future work needs to focus on adapting the model architecture to handle varying well-log measurement types and improving interpretability of feature interactions. The method shows promise for broader applications in subsurface characterization and automated geological analysis.

Supplemental Information

Supplemental Information 1 Code.

Additional Information and Declarations

Competing Interests

An Hai Nguyen is employed by Petrovietnam Exploration Production Corporation T. Khang Nguyen is employed by IBM Vietnam. The authors declare that they have no competing interests.

Author Contributions

An Hai Nguyen conceived and designed the experiments, performed the experiments, analyzed the data, performed the computation work, prepared figures and/or tables, authored or reviewed drafts of the article, and approved the final draft.

Khang Nguyen conceived and designed the experiments, performed the experiments, analyzed the data, performed the computation work, authored or reviewed drafts of the article, and approved the final draft.

Nga Mai analyzed the data, authored or reviewed drafts of the article, and approved the final draft.

Data Availability

The following information was supplied regarding data availability:

Data from Hall (2016) are available at GitHub: https://github.com/mardani72/Facies-Classification-Machine-Learning

Code is available in the Supplemental Files.

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
