# Peer review of "Classifying reservoir facies using attention-based residual neural networks"

_PeerJ Computer Science, doi:10.7717/peerj-cs.2977_

## Round 0.1 · original submission · Major Revisions

Thank you for submitting your manuscript to our journal. While the reviewers recognize the potential significance of your research, they have identified substantial concerns that require extensive revisions before further consideration for publication. The reviewers have also highlighted significant gaps in their comments. These concerns need to be addressed. Given the extent of these revisions, we invite you to submit a thoroughly revised manuscript, accompanied by a detailed response addressing each point raised by the reviewers. If you choose to resubmit, your manuscript will undergo another round of review to assess how effectively the concerns have been addressed.

Reviewer 1 ·

Basic reporting

The manuscript presents a novel approach for classifying reservoir facies using an attention-based residual neural network. The proposed framework integrates Transformer-based attention mechanisms with deep residual networks to handle both categorical and numerical well-log features effectively. The study is evaluated using well-log data from eight wells across diverse geological settings, demonstrating superior classification performance compared to traditional machine learning and state-of-the-art deep learning methods. Notably, the model achieves AUROC and AUPRC values of 0.883 and 0.502, respectively, marking a significant improvement over existing techniques. The paper highlights the importance of feature fusion, the stability of the proposed architecture, and its reproducibility across multiple implementations. Despite these strengths, the paper has several areas for improvement. The methodology could provide more details on the hyperparameter selection process and training dynamics. Additionally, the experimental section lacks discussions on potential biases in the dataset and the impact of different well-log features on the model's performance. Furthermore, while the proposed model demonstrates strong results, an ablation study on individual components (e.g., the effect of the attention mechanism versus residual connections) would further substantiate the claims. Finally, the discussion on model limitations and computational overhead should be expanded to provide a balanced assessment of its applicability in real-world scenarios.
1) First of all, related to the contents within the abstract, authors are recommended to provide a clearer statement on the novelty of their approach compared to existing deep learning methods. Specifically, emphasizing the unique integration of attention mechanisms and residual connections would strengthen the contribution claim. Additionally, the performance improvement compared to baselines should be contextualized by explaining how these improvements translate to real-world applications.
2) Additionally, within the introduction of this paper, the authors should expand on the motivation behind combining Transformers and residual networks. While the introduction discusses the challenges of facies classification, it would benefit from a more detailed justification of why attention mechanisms are particularly suitable for well-log data. Furthermore, a discussion on the interpretability of the model in a geoscience context would enhance its relevance to domain experts.
3) For the literature review of this paper, authors are recommended to provide a clearer comparison between traditional machine learning approaches and recent deep learning advancements. While various ML and DL techniques are mentioned, a more structured synthesis of their limitations would better highlight the research gap addressed by this work. Additionally, citing recent studies that have employed similar hybrid architectures would strengthen the contextual background.
4) Related to the methodology of this paper, authors are suggested to improve clarity and technical rigor, such as the explanation of the Transformer block should include a brief mathematical formulation of how attention mechanisms improve feature learning in well-log data. Moreover, authors should also add more details on the residual block implementation, including justification for the chosen activation functions and normalization techniques, which would enhance reproducibility.
5) Additionally, the model training procedure should specify hyperparameter tuning strategies (e.g., learning rate scheduling, batch size selection, and regularization techniques). While the dataset description is thorough, a discussion on data preprocessing steps, such as handling missing values or feature engineering, would be beneficial.
6) Finally, an ablation study investigating the impact of key architectural components (e.g., attention layers, residual connections) should be included to validate the necessity of each design choice.

Experimental design

Please refer to my "Basic reporting" section.

Validity of the findings

Please refer to my "Basic reporting" section.

Additional comments

No comment.

Cite this review as

Reviewer 2 ·

Basic reporting

The manuscript demonstrates a clear structure and professional English. The literature review provides comprehensive information and thorough evaluations that effectively address the topic of the field. The figures and tables are well-prepared and clearly presented. However, several aspects of the manuscript would benefit from further development and clarification as detailed below.

Experimental design

The experimental methodology requires additional changes in:

1. Methodological Detail: The manuscript would benefit from a more comprehensive description of the experimental procedure, including specific approaches to data preprocessing, model selection criteria, and hyperparameter tuning. Highlighting method details would strengthen the reader's understanding of this research.

2. Data Visualization: Given that the dataset incorporates both categorical and numerical features, visual representations of the data distributions and relationships would substantially enhance the reader's understanding. We recommend including appropriate exploratory data analysis (EDA) visualizations such as histograms, box plots, and correlation matrices to illuminate the dataset characteristics.

3. Performance Metrics: The evaluation of this study should be significantly enriched by incorporating more metrics to provide readers with a holistic assessment of model effectiveness. Authors are recommended to report other metrics including Accuracy, Precision, Recall, and F1-score.

4. Data Sampling: While the manuscript mentions a 9:1 ratio for training and validation data splitting, it should specify whether this sampling was implemented through random or stratified sampling. This distinction is crucial as it directly impacts the balance of class distributions across partitions and consequently affects the validity of the results. Please provide a clear rationale for the chosen approach.

Validity of the findings

The comparative analysis and discussion components require further development:

1. Comparative Framework: While the manuscript includes comparisons with certain deep learning models, expanding the comparative framework can improve the study's contribution. Authors are recommended to implement modern deep learning architectures for tabular data (e.g., TabTransformer, TabFPN) to provide a fair comparison.

2. Analytical Depth: Several sections may need to improve with deeper analysis. Besides, the discussion part should explain the causal factors underlying performance difference between models.

Cite this review as

Reviewer 3 ·

Basic reporting

This manuscript presents a method for classifying facies using an attention-based deep learning model. This work is well-explained with sufficient details in methodology, related works, and experiments.

Experimental design

Although this research has merits for publication, some minor points need to be resolved.
(a) The methodology section lacks sufficient details. Please provide a more comprehensive description of the approach, including specific steps in data preprocessing, feature selection, and model training.

(b) The paper should include comparisons with other relevant methods to strengthen the validity of the findings. Specifically, consider comparing with more traditional machine learning models such as Decision Tree and CatBoost.

(c) The paper does not discuss the applicability domain of the proposed method. Please elaborate on the scope and limitations of the approach, including potential constraints when applying it to different datasets or problem settings.

(d) To further validate the robustness of the proposed method, testing on an external dataset is recommended. This would provide a clearer understanding of the model's generalizability beyond the current dataset.

Validity of the findings

This work provided sufficient statistical evidence proving model stability and reproducibility

Cite this review as

---

## Round 0.2 · Minor Revisions

Thank you for your submission, which presents interesting findings in this field. The reviewers find merit in your work and believe it has the potential to make a valuable contribution after addressing several specific issues. We recommend minor revisions to enhance clarity and strengthen your manuscript. The reviewers have also highlighted several minor gaps in their comments. These concerns need to be addressed. Given the extent of these revisions, we invite you to submit a revised manuscript, accompanied by a detailed response addressing each point raised by the reviewers. If you choose to resubmit, your manuscript will undergo another round of review to assess how effectively the concerns have been addressed.

Reviewer 1 ·

Basic reporting

After carefully reviewing the revisions as well as feedback of authors within the latest submitted version of this paper, I confirmed that all problems within previous paper have been sufficiently addressed, as a result I thought this paper could be accepted for publication in this form. Thanks.

Experimental design

No comment.

Validity of the findings

No comment.

Additional comments

No comment.

Cite this review as

Reviewer 2 ·

Basic reporting

- The manuscript is clear and well-structured, but some parts still need small improvements noted below.
- In Figure 2, the label for 'Facies' should be aligned consistently with the other measurement labels.
- In Table 4 and Table 5, please indicate which model achieves the best score for each metric by bolding the top values to aid quick comparison.

Experimental design

- The authors have added a more detailed description of methodological aspects, including model selection criteria, and comprehensive hyperparameter tuning.
- The method provides a sufficient explanation of the data processing steps and model design.

Validity of the findings

- The previous suggestion has been addressed, including information on analytical depth and comparative framework
- The conclusions are well supported by the thorough experimental results, showing clear improvements over baseline and competing methods.

Cite this review as

---

## Round 0.3 · Minor Revisions

Based on reviewers' decisions, we are delighted to inform you that your manuscript, "Classifying reservoir facies using attention-based residual neural networks", is almost ready to be accepted for publication in PeerJ Computer Science.

Firtst, please address the following from the Section Editor:

> Although the current paper separates background, related works, and motivation, a more integrated 'Introduction' section may improve narrative coherence and better align with standard article formats. I recommend merging these sections into a unified introduction with logical flow from problem definition to contribution. So the introduction section in its current form require revision for clarity, conciseness, and structural coherence.

Reviewer 1 ·

Basic reporting

After carefully reviewing the revisions as well as feedback of authors within the latest submitted version of this paper, I confirmed that all problems within previous paper have been sufficiently addressed, as a result I thought this paper could be accepted for publication in this form. Thanks.

Experimental design

Please refer to the contents in my "Basic reporting" section.

Validity of the findings

Please refer to the contents in my "Basic reporting" section.

Additional comments

No comment.

Cite this review as

Reviewer 2 ·

Basic reporting

- The manuscript is now clearly written and well-organized. The authors have addressed previous minor issues, including label alignment in Figure 2 and formatting in Tables 4 and 5. These changes improve the clarity and readability of the presentation.

Experimental design

- The methodological description is now more complete, with clear justification for model selection and hyperparameter tuning. The workflow from data processing to model evaluation is well explained and appropriate for the study objectives.

Validity of the findings

- The authors have provided adequate experimental evidence to support their conclusions. The revised analysis and comparison framework improve the reliability of the findings, demonstrating the proposed method's effectiveness.

Cite this review as

Reviewer 3 ·

Basic reporting

no comment

Experimental design

no comment

Validity of the findings

no comment

Cite this review as

---

## Round 0.4 · accepted · Accept

We are delighted to inform you that your manuscript, "Classifying reservoir facies using attention-based residual neural networks", has been accepted for publication in PeerJ Computer Science.